# Unpacking privacy: Valuation of personal data protection

**Anya Skatova**[1,2,3,4]*, **Rebecca McDonald**[2,5], **Sinong Ma**[3,6], **Carsten Maple**[3,4]

**1** Bristol Medical School/School of Psychological Science, University of Bristol, Bristol, United Kingdom, **2** Warwick Business School, University of Warwick, Coventry, United Kingdom, **3** Warwick Manufacturing Group, University of Warwick, Coventry, United Kingdom, **4** Alan Turing Institute, London, United Kingdom, **5** School of Economics, University of Birmingham, Birmingham, United Kingdom, **6** Queen's Management School, Queen's University Belfast, Belfast, United Kingdom

* anya.skatova@bristol.ac.uk

## Abstract

Information about individual behaviour is collected regularly by organisations. This information has value to businesses, the government and third parties. It is not clear what value this personal data has to consumers themselves. Much of the modern economy is predicated on people sharing personal data, however if individuals value their privacy, they may choose to withhold this data unless the perceived benefits of sharing outweigh the perceived value of keeping the data private. One technique to assess how much individuals value their privacy is to ask them whether they might be willing to pay for an otherwise free service if paying allowed them to avoid sharing personal data. Our research extends previous work on factors affecting individuals' decisions about whether to share personal data. We take an experimental approach and focus on whether consumers place a positive value on protecting their data by examining their willingness to share personal data in a variety of data sharing environments. Using five evaluation techniques, we systematically investigate whether members of the public value keeping their personal data private. We show that the extent to which participants value protecting their information differs by data type, suggesting there is no simple function to assign a value for individual privacy. The majority of participants displayed remarkable consistency in their rankings of the importance of different types of data through a variety of elicitation procedures, a finding consistent with the existence of stable individual privacy preferences in protecting personal data. We discuss our findings in the context of research on the value of privacy and privacy preferences.

## Introduction

With the rising prominence of digital technology in the modern economy, the amount of data recorded about consumers' behaviour and retained in various virtual and physical places is enormous. This personal data can reveal private details about consumers' everyday behaviour. For example, records of individuals' daily journeys are reflected in mobile phone or car GPS data; household composition can be deduced from electricity use in the home; loyalty card data can reflect personal lifestyle choices; and bank transaction data can reveal details of

**Data Availability Statement:** https://osf.io/s35ur/.

**Funding:** AS, RM - ESRC ES/K002201/1 https://www.ukri.org/councils/esrc/ AS, SM, CM - EPSRC EP/N02334X/1 https://www.ukri.org/councils/epsrc/ The funders had no role in study design,

data collection and analysis, decision to publish, or preparation of the manuscript.

**Competing interests:** The authors have declared that no competing interests exist.

personal financial circumstances. These opportunities are increasingly being exploited in new and innovative ways: personal data has economic value for government organisations and businesses [e.g., in advertising, 1]. It enables a diverse collection of digital economy services to be provided, serving a foundation for many successful business models [2; see 3 for a review of the role of personal data in a variety of business models]. In this paper, we aim to understand preferences for protecting personal data.

## Value of personal data and preference for privacy

Economists have long been interested in how personal information is shared and, broadly speaking, how it is valued (by Acquisti et al. [3], see also Stigler [4] and Poster [5]). This work brings into focus consumers' views on the consequences of their privacy-related behaviours and measures their preferences regarding privacy and value of personal data [e.g., 6–12]. An individual deciding to share data online may be compromising their privacy, since this data can reveal private information about them. Therefore, an important factor influencing an individual's preference for protecting their personal data is how much they value their privacy, captured by their preference for privacy. Privacy preferences are a feature of some theoretical models of online privacy including Montes et al. [8]. Relatedly, the well-established idea of the "privacy calculus" [13] is that individuals assess the costs and benefits of protecting (or sharing) their information—essentially asking themselves whether it is "worth it" to reveal a certain type of information. To engage in such a calculus, individuals must have defined preferences over their privacy. However, some studies raise doubts about the existence of such preferences [e.g. 14]. We review the literature regarding the existence and nature of preferences for privacy, and the value of privacy, in what follows.

An early empirical investigation into the value of privacy was undertaken by Hann et al. [15], who conducted a Discrete Choice Experiment to investigate the trade-offs between privacy and benefit (financial and time saving) in the choice between two websites. The authors demonstrated that their sample of students were willing to trade off their privacy for sufficient economic benefit. Huberman et al. [16] explored individuals' willingness to accept compensation for revealing sensitive personal information, specifically one's bodyweight. They showed that people were willing to engage in such a trade, but that those who perceived their weight to be different from the norm demanded greater compensation for sharing this data. This again suggests that individuals are entering into a form of privacy calculus.

## Willingness to pay to protect different types of personal data

Since these early investigations, there has been ongoing interest in attempting to value the protection of privacy online [e.g., 17]. A systematic review of the literature regarding "putting a price tag on personal information" was conducted by Wagner et al. [18]. They found that the type of information being disclosed (or protected) was an important feature in determining the amount that individuals were willing to pay (or accept). The second feature that they discovered to be important was the privacy impact of personal information disclosure. Specifically, they found that:

> "the more sensitive the data and the more identifiable people are, the higher has been the price people attach to their data as they perceive higher risks" [18, p 3766].

As we have seen, there is substantial variation in the evidence about willingness to pay for protection of personal data. This may reflect the lack of any clear understanding amongst the participants—and arguably amongst the researchers—about exactly what they mean by

privacy. Many of the articles refer to the value of privacy, but actually test willingness to pay not to share (or, willingness to accept compensation for sharing) a particular type of personal data [e.g. 19 for the results of a logic test; 20 for social media details]. This means that the value attributed to privacy will be confounded by all the other reasons that someone may wish to avoid sharing that particular type of personal data. Our view is that privacy relates directly to what sharing a piece of data can reveal about the person's behaviour. This will differ on many dimensions, including context, who is requesting the data, and on the specific data type. As such, to understand how individuals value their privacy, a first step is to consider their willingness to share a variety of different types of personal data.

However, while the values of different data types might vary, there still could be underlying individual privacy preferences. Many of the studies to date that explore the value of privacy (through eliciting the value of personal data) focus on just one type of data [e.g., demographic information, 11]. In this case, the consistency and stability of preferences for protecting personal data cannot be investigated, and the robustness of any conclusions about how much individuals value personal data in general cannot be ascertained. Exceptions to this mainly investigate the comparison between willingness to accept and willingness to pay [e.g., 12] or between different types of willingness to pay [e.g., 10] but still with the same pieces of personal information. However, different types of data—for example energy use at home versus bank transactions—can reveal different information about an individual, and these different revelations may be sensitive to different extents. There is evidence that people care about some types of data more than others, and that people differ with respect to which types of data they care about [21, 22]. Arguably, then, the value of personal data is not a unified concept with a single value but is context-dependent and will vary by different types of data [3]. Further, consumers with the same underlying privacy preferences might make different decisions while sharing different types of data because the assessment of risks and benefits will be different in each data sharing environment.

To summarise this literature, it appears that individuals are often willing to give up privacy for benefits defined in terms of money and/or convenience, and that their willingness to engage in such trade-offs may differ between contexts where the risks and benefits of the trade-offs vary. This evidence is consistent with the idea that privacy preferences exist, that they are a component of the value of personal data, and that they may influence the decision whether to share personal data online, alongside other influences related to the decision context.

However, an omission in the current literature is evidence about the stability and consistency with which individuals can express their privacy preferences. If individuals' stated preferences regarding their personal data are not consistent, stable and well-defined, then behaviourally-relevant models, in which online behaviour is governed in part by privacy preferences, will be intractable. Our paper aims to understand preferences for protecting personal data through assessing individuals' rankings of the importance of protecting different types of personal data, including (but not limited to) analysing their willingness to pay to protect them. We next describe in more detail existing methodological work on the stability of privacy preferences, and draw on related literature in other contexts, to ascertain how best to capture the stability—or otherwise—of preferences.

## Preference elicitation across contexts

To understand whether people can consistently report their preferences, it is necessary to elicit these preferences in different ways. To our knowledge just one paper has compared different methods of preference elicitation in the context of online privacy. Benndorf and Normann

[10] investigated the effect of two different methods in eliciting the value of personal information, comparing a Becker-DeGroot-Marshak style second price auction [23] with a Take it or Leave it approach. They found a significant difference in values between these approaches, even though both aimed to elicit the same underlying construct. Hence, they provided preliminary evidence that preferences to keep personal data private—at least in terms of their monetary valuations—are subject to elicitation method effects. To our knowledge no research has yet considered the (in)consistency with which individuals express their preference ranking over the importance of protecting different types of personal data, and whether those rankings depend on elicitation methods. However, evidence about the stability and reliability of preferences expressed through different elicitation procedures has been considered in detail in other fields, most notably in experimental economics and in the non-market valuation literature applied to health outcomes.

Methods that exist to categorise and compare the strength of preference over different outcomes come in many forms, from simple choice tasks through to complex trade-off tasks between the outcome and some numeraire of its value [for reviews comparing methods in the health context, see 24; 25]. One important way to categorise these methods is by their relative or absolute nature. Relative tasks allow a researcher to elicit the order—and sometimes the strength of preference—for one outcome compared to another. Examples include direct ranking tasks, where a participant must rank outcomes in order of preference, and binary choice tasks where one option is compared to another and the preferred option is chosen. Absolute tasks require participants to reveal the strength of preference for an outcome by itself without reference to any alternatives. For example, with rating scales an outcome is rated in terms of its quality, likeability or importance; while in valuation tasks an outcome is judged according to its monetary value.

According to standard economic theory, as long as individuals hold well-defined and consistent preferences over the outcomes in question, the type of task used to elicit these preferences should not influence the outcome of the preference elicitation. Put simply, if I like an apple more than an orange, then I would choose an apple over an orange in direct choice, and I would be willing to pay more for the apple than I would be willing to pay for the orange. However, a long and rich literature in experimental economics and psychology has revealed that choice and judgement tasks follow different psychological processes [e.g. 26] and prioritise different attributes of a comparison [e.g. 27]. It is therefore an open question whether individuals can consistently reveal their preference ordering with respect to the importance of protecting different types of their personal data. Arguably, the more well-defined their underlying preferences, the more stable the preference ordering will be when compared across different elicitation tasks.

The health economics literature provides further evidence to support the notion that preference elicitation procedures matter. Discrepancies between values of a health outcome have been found depending on the method used to elicit them [e.g. 24, 28–30]. There is a wide array of reasons given for these discrepancies, ranging from anchoring [e.g. 31] to reference point effects and coherent arbitrariness [32]. Taken to the extreme, some authors have argued that there may not even be such a thing as underlying, stable and coherent preferences [see for example 33–35 for various discussions along these lines]. There are striking similarities between valuing and ranking the importance of different health states, and valuing and ranking the importance of protecting different types of personal data. Both health states and personal data scenarios involve making judgements and choices that involve negative emotions like fear and distress. Both scenarios are relatively unfamiliar, in the sense that few people are regularly asked to directly evaluate different scenarios regarding their health, nor their privacy. Finally, both scenarios are difficult, arguably impossible, to incentivise, yet have important

implications for regulation and policymaking. The UK Treasury's Green Book guidance for policy appraisals states that

> "If robust revealed preference data is not available, surveys that use willingness to pay and willingness to accept are an established alternative method known as stated preference techniques" [2, p. 42].

It seems wise, then, to draw on the lessons from health economics when considering the value of personal data and preferences to keep the data private. We put the methods used in the health economics literature to work in the context of privacy valuation, checking for the implicit ranking of types of data that they imply, and inferring the plausibility of using these elicitation techniques to establish the value of personal data to members of the public. We focus on the comparison between rankings of types of data that are based on absolute judgment (willingness to pay) tasks, and rankings of types of data based on relative judgement methods, to understand the extent to which preference orderings for personal data are stable across these frames.

We emphasise that we are not advocating for a market for personal data in which service users are forced to pay for online privacy. The ethical implications of this possibility are complex and their detailed exploration is outside the scope of this research. Instead, and following the rich traditions of non-market valuation in other fields like health and the environment, we are simply using Willingness to Pay as a tool for establishing the value that individuals place on their personal data. In principle, this information can feed into regulatory processes, and in particular can provide an estimate of the benefits of privacy protection measures in cost-benefit analyses. In our setting, the values are captured to allow us to explore the stability of the expressed preferences, given that those preferences are not directly observable.

In this study, **we aimed to understand preferences for protecting personal data**. Instead of assuming preferences for privacy to be captured by an exogenously determined "privacy value" constant in the utility function in all contexts, we explored how preferences differ across different data sharing environments. We did not assume that preferences are well-defined and stable but used a multitude of elicitation techniques to explore the consistency with which different types of data are ranked and valued.

Clearly, privacy is a complex issue that encompasses, and feeds into, many other related concepts. We adopt a simple conceptual approach (illustrated in Fig 1) in which we assume that preferences for (online) privacy are a contributor to preferences for avoiding sharing personal data, but they not the whole story. The context of the data sharing scenario will also drive a preference for avoiding sharing personal data, but is not itself part of a preference for privacy. We further consider that Willingness to Accept (or Pay) for sharing (or not sharing) personal data online is driven in part by the preference for avoiding sharing personal data, and hence by preference for privacy, but that again other considerations may play a role, such as ability to pay. At the heart of the framework is the underlying preference for privacy, and the stability of these underlying privacy preferences are necessary (although not sufficient) for the consistency of decisions about sharing personal data, and this is what we explore in our empirical work.

We took a neutral approach when defining the context in which participants were asked to report the strength of preferences for keeping their personal data private, and we did not define the specific intended use of the information they were asked to share. Our study mimics real world online environments where individuals need to make their own assessment of costs and benefits associated with sharing of a particular type of personal data, and judge for themselves whether the trade-off is worth making.

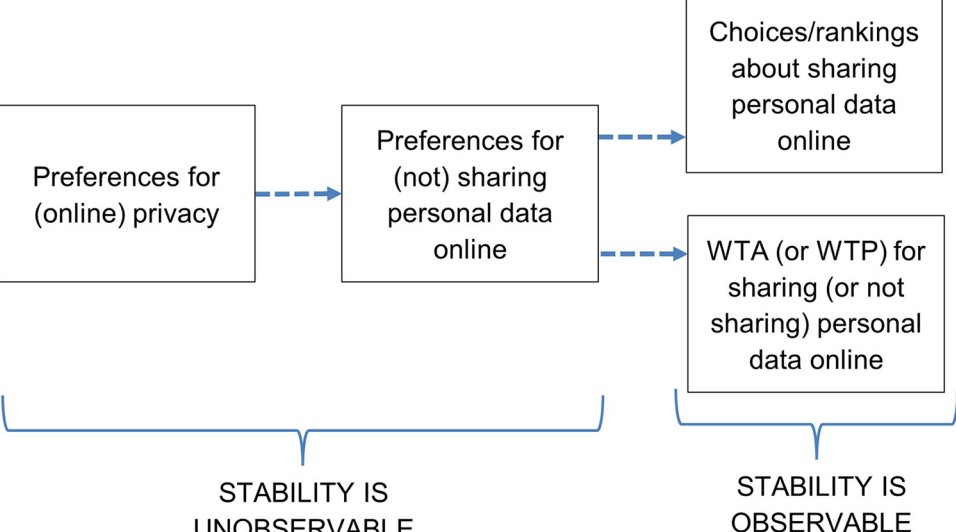

**Fig 1. Framework for considering privacy preferences, sharing preferences, and related behaviour.** Dotted arrows indicate that the components are linked, but that other influences (such as context) also contribute to determining the next component.

Our approach does not intend to give a detailed exploration of the reasons behind the decision whether or not to share personal data, but instead focuses on examining two main issues: how individuals value keeping personal data private in a variety of contexts and how consistently they report these preferences. We explored the reliability and stability of these preferences taking a holistic view of the prospect of sharing personal data in the context of the digital economy and compared individuals' overall willingness to share their data across personal data environments and elicited through different methods.

Our research extends previous work on factors affecting individuals' decisions about whether to share personal data. We take an experimental approach and focus on whether consumers place a positive value on protecting their data by examining their willingness to share personal data in a variety of data sharing environments, including their Willingness To Pay (WTP) to protect it. Using five evaluation techniques, we systematically investigate whether members of the public value keeping their personal data private. Specifically, we determine (1) whether stated preferences for keeping personal data private are stable within individuals; and (2) whether stated preferences for keeping personal data private systematically vary between data sharing environments (e.g., electricity consumption vs physical activity monitoring vs spending patterns).

We find that 96% of individuals are willing to pay to avoid sharing their personal data in at least one of data sharing environments, indicating that privacy is valuable. We also find that individuals are consistent in their rankings of the importance of different types of personal data when these rankings are elicited in different ways. This is evidence consistent with the notion that the techniques we employ can reveal stable underlying preferences for keeping personal data private. We find consistent differences across our sample in their stated preferences for keeping different types of personal data private: financial and medical data were typically rated as the most important to protect, while supermarket loyalty cards, electricity use and physical activity data were rated as the least important to protect. We conclude that stability in stated preferences for keeping personal data private across a variety of elicitation methods is evidence that privacy preferences are well defined. If so, the stated preferences we elicit may

help to explain how individuals may decide whether to share their personal data in the context of the digital economy. Finally, we draw conclusions on the links between our findings on willingness to pay to protect personal data in different environments with the literature on the value of privacy.

## Experiment

This paper investigates the stability of participants' preferences for keeping their personal data private. In an online experiment, we asked whether participants place a positive monetary value on protecting different items of their personal data and explored how their willingness to pay differed between different data sharing environments. We drew inspiration from non-market valuation economic methodologies [e.g., contingent valuation, with its origins in 36; rating scales as explored in 31; paired comparison, which draws on insights from 37], designing tasks in which participants expressed their absolute and relative judgments about the importance of protecting different items of their personal data. Through these tasks we could establish whether and how people value the privacy of their online data, whether they value different types of data differently and whether there is consistency between the rankings revealed by different elicitation methods.

To achieve the overall aim of understanding privacy preferences, the specific goals of our research were twofold. First, we aimed to ascertain the relative importance of each data type, as well as preliminary evidence about the levels of willingness to pay (WTP) for keeping each data type private. Second, we aimed to explore the consistency of the rankings of the data types across elicitation methods. To address these questions, we elicited the relative importance of protecting different personal data types using five different elicitation mechanisms, and gathered demographic information including age, gender, education and income to investigate whether WTP is affected by any of those factors. Two elicitation methods focused on respondents' WTP for keeping their personal data private, one further method focused on the perceived absolute importance of personal data types and two focused on respondents' ranking of the importance of protecting different types of personal data—one pairwise ranking and one direct overall ranking. This is, to our knowledge, the first study that directly addresses the effect and consistency of different elicitation mechanisms in the context of data protection.

**Participants & procedure.** A total of 265 participants took part in the online study run on the Qualtrics platform. The study took up to 30 minutes and participants received a fixed payment of £3 in return. The study was approved by Humanities and Social Sciences Research Ethics Sub-Committee, University of Warwick. The dataset is available upon request from the first author. Full experimental design and exact questions can be found in the S1 File.

The survey was distributed through Prolific Academic (Prolific.ac). Three participants failed to answer all questions, and 262 responses were used in all of the following analyses. Of the respondents, 69.85% were female. Participants reported an average age of 37.28 (SD = 11.42), ranging from 18 to 72. Regarding education level, 53.44% reported no undergraduate degree or equivalent, 35.11% reported an undergraduate degree or equivalent, and 11.45% reported a degree higher than undergraduate. Our sample was more educated and contained a marginally higher proportion of female participants than the general population, since the UK population is 50.89% female [38], and only 27.22% of the population have an undergraduate degree or higher qualification [39]. Income was measured on an interval scale with ten intervals: "less than £10,000"; "£10,000 to £19,999"; "£20,000 to £29,999"; "£30,000 to £39,999"; "£40,000 to £49,999"; "£50,000 to £74,999"; "£75,000 to £99,999"; "£100,000 to £149,999"; "£150,000 to £199,999"; and "£200,000 or more". There was also an option not to respond. The midpoint of each income interval was used as a proxy for the participant's

income. The median of the household's annual income before tax for our sample was £25,000, equivalent to around $33,700 (SD = £23,600 or $32,000), which is lower than that of UK population in general: the Office for National Statistics [40] reported median UK household pre-tax income of £33,705 for 2017. Income levels for people who preferred not to reveal this information (12 participants) were treated as missing values.

**Experimental design.** The main preference elicitation section of the experimental design consisted of five conditions: two willingness to pay conditions, as well as three additional conditions to elicit individual preferences to protect different types of data. Screen shots of the five conditions are provided in Fig 2 and the full experimental protocol is available in S1 File. Every participant completed all five conditions, and the conditions were presented in a random order. The responses in the WTP conditions revealed whether the participant was willing to pay anything at all to protect the data types, and if so, how much they were willing to pay. The responses in the three non-WTP conditions revealed participants' ranking of data types in terms of their perceived importance of protecting the data. This allowed us to test the consistency of subjective rankings compared to the rankings implied by the WTP conditions.

All conditions had a common scenario. Participants were required to imagine that they had just bought a new smartphone. In addition to having all the usual functions, it also had a special application which intended to assist them in their everyday life by tracking different types of their personal data. It was emphasised that the use of this application was very desirable for a participant. The scenario specified that when they used the personal data application, they

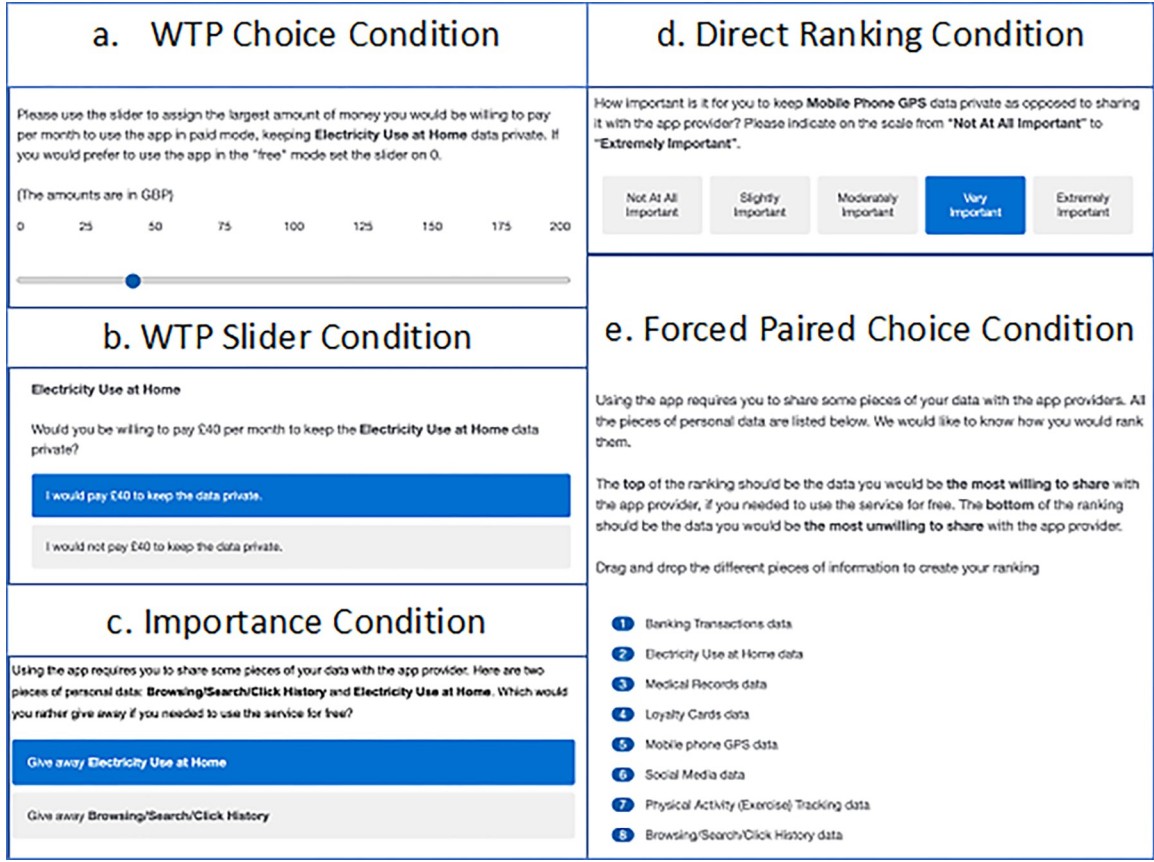

**Fig 2. Each sub-section displays a screenshot of one of the five experimental conditions.** The order of conditions was randomised between participants.

were given two choices; they could run the app in "free" mode, or they could run it in "paid" mode. In "free" mode, they did not have to pay anything but the company providing the app collected information about them when they used the app and used that information for various marketing and other purposes. In "paid" mode, they would pay a specified amount of money for the service and their data would remain private. In "paid" mode, the data would remain anonymous in a database and would only be used to maintain the functionality of the app. We did not specify the purposes for which the data might be used in "free" mode, allowing individuals to make up their own minds about the likely uses and value of their personal data for companies.

The personal data, defined here as "*D*", could be one of the following: Banking Transactions, Browsing/Search/Click History, Electricity Use at Home, Loyalty Cards, Medical Records, Mobile Phone GPS, Physical Activity (exercise) Tracking and Social Media. These eight types of data were selected from a larger set of data types based on previous research (Skatova et al. 2013) as well as because they are commonly used in real life and are familiar to the general public.

**WTP choice condition** Participants were first asked whether they were willing to pay anything, even a small amount, to keep the data *D* private, or alternatively whether they would prefer to use the app for free and share their data *D*. If they preferred to pay, we asked them whether they would be willing to pay Great British Pounds (£) 40 per month to keep the data private or else use the app for free and share the data. Based on their response, different amounts of money were presented, and respondents had to accept or reject paying that amount to keep the data private.

The amounts of money were iterated between the bounds £0.01 and £200 until the person was indifferent. The iteration procedure was designed to reveal bounds on the individual's indifference point, which was then coded as the midpoint of these bounds. For instance, if an individual sequentially confirmed they would be willing to pay £40, £125 and £175 per month to keep data *D* private but they would not be willing to pay £200, then their highest acceptable amount is £175 per month while their lowest unacceptable amount is £200 per month. This individual's WTP would be recorded as £187.50 per month. If a participant indicated that they would be willing to pay £200, their indifference point could not be determined by the iteration. There were ten cases where participants were willing to pay more than £200 to protect some of their data types. In these cases, we asked them to state the maximum amount they would be willing to pay to keep this specific type of data private, which ranged from £210 to £300. Participants completed this procedure eight times, once per data type. The starting value of £40 and the maximum value of £200 was chosen based on pilot work with an open-ended WTP task.

**WTP Slider condition** Participants used a slider to directly assign the largest amount of money they would be prepared to pay to keep personal data *D* private. The slider ranged from £0 to £200 per month. The slider was set at a default of £40 per month to start with. This amount, and the £200 maximum, were chosen to match the £40 starting amount in the WTP Choice condition. Participants answered eight WTP slider tasks, one per data type.

**Importance condition** Participants used a five-point scale, ranging from "not important at all" to "very important", to rate the importance of keeping each piece of personal data *D* private as opposed to sharing it. Participants answered eight of these tasks, one per data type.

**Direct Ranking condition** Participants ranked the 8 types of personal data in order of the importance of keeping them private. They had to drag and drop types of data in a vertical ranking, where the top of the ranking was the data they were most willing to share, if they needed to use the app for free. At the bottom of the ranking was the data they were the most unwilling to share. Participants completed one ranking which included all eight data types.

**Forced Paired Choice condition** Participants made a series of forced paired choices.

Each choice was between two pieces of personal data, data $D_1$ and data $D_2$. Each time, they needed to decide which data type they would prefer to give away if they needed to use the app for free. Participants completed 28 of these choices, one for every unique binary pairing of the data types.

**Analytic approach.** We first tested whether participants were consistent in their privacy preferences by calculating how consistent the rankings of data types were between the five conditions within each participant. We then used a probit and an OLS regression to analyse whether the demographic information explained variation in whether and how much individuals were WTP to keep their data private, whether WTP depended on the types of data, and whether it depended on within-person consistency scores.

*Individual consistency in ranking different data types.* To examine consistency in individual preferences for sharing different types of data, we converted people's responses in each condition into rankings. The *Direct Ranking* condition data was used without conversion. For the *WTP Choice* and *WTP Slider* conditions we assumed that a higher WTP for a data type indicated a lower willingness to share. For the *Importance* condition we assumed higher-rated importance indicated a lower willingness to share. For the *Forced Paired Choice* condition we counted the number of times each data type was chosen as the one to share, and assumed that a lower count implied lower willingness to share the data. This way, if banking data was never given away it would have the lowest willingness to share and thus the highest importance rank, and if loyalty cards data was always given away, it would have the highest willingness to share and the lowest importance rank. For any ties in rankings, standard competition ranking was employed to determine the Willingness to Share (WTS) rank. This was converted to an importance score where the number increases with importance of the data type. S1 Appendix gives an example.

To measure consistency of each individual's ranking across different conditions we adopted Krippendorff's coefficient [41].

The alpha coefficient is defined as [41, page 222]:

$$a = 1 - D_0/D_e$$

where $D_0$ is a measure of the observed disagreement and $D_e$ is a measure of the disagreement that can be expected by chance. In our context, the observed disagreement refers to the individual participants inconsistency in rankings for the same data type across different conditions. The coefficient ranges from -1 to 1: when individuals' sensitivity rankings are completely consistent across elicitation methods and inconsistent choices are completely absent, $D_0 = 0$ and $a = 1$. When consistent choice and inconsistent choices are purely driven by chance, $D_0 = D_e$ and $a = 0$. Alpha less than zero implies systematic differences in ratings that are not due to random chance, for example if two methods elicited opposite ratings for the data types.

*Regressions.* To estimate whether demographic and control variables influenced the likelihood of being willing to pay something greater than zero to protect personal data, we used mixed-model probit regressions with a random effect to account for the fact that each participant provided eight answers in each of the two WTP conditions. The Banking Transactions data type was used as a reference category. The following model was specified:

$$Prob\ (y_i = 1) = F(\beta_0 + \beta_1 T_i + \beta_2 K_i + D_i\gamma + \varepsilon_i)$$

where $Y_i$ is a binary variable taking the value 1 if the individual chose to pay money to protect the data, and 0 otherwise; $D_i$' is a vector of demographic variables (age, gender, education and

income)*; $K_i$ is the individual Krippendorff's alpha coefficient [41] and $T_i$ is the type of data. The function $F$ is the cumulative distribution function of the standard normal distribution.

To estimate whether demographic and control variables influenced *the level of WTP to keep data private* we used mixed-model ordinary least squares (OLS) regressions, again with a random effect of a participant [e.g. 42, 43]:

$$Y_i = \beta_0 + \beta_1 T_i + \beta_2 K_i + D_i'\gamma + P_i'\delta + \varepsilon_i$$

where $Y_i$ is an individual's WTP for keep their data private and is treated as continuous; $D_i'$, $T_i$, $K_i$, are defined as in the probit model.

## Results

### Are people consistent in their judgements about different types of personal data between conditions?

We calculated average rankings for each data type in each condition, pooling across participants. The aggregate ranking of each data type between different conditions was similar, with the exception of Mobile Phone GPS: see Fig 3 for aggregate ranking scores and corresponding 95% Confidence Intervals, and Fig 4A–4C for density distributions of ranks for each data type and each condition.

Fig 3 demonstrates three clear "tiers" of data. Banking Transactions and Medical Records, which we determine to be "Tier 1" data, are ranked consistently as the most valuable between all conditions, and ranked on average as 7.52 and 6.98, out of 8, respectively. A second tier of data types includes Browsing History, Mobile Phone GPS and Social Media Data, which are consistently ranked just under 6, with the exception of a lower ranking implied by the Forced Paired Choice condition for all three data types, and higher ranking for Mobile Phone GPS data in the WTP Choice condition. Finally, the lowest importance category, "Tier 3", of data includes Electricity Use, Loyalty Cards and Physical Activity data, rankings for which vary between 2 and 5 across all conditions. On average, the ranking of different data types appears consistent between conditions, with the exception of Mobile Phone GPS which is ranked in the first tier in WTP Choice but not in other conditions.

The Fig 4 show that Tier 1 data types are the most consistently ranked between different conditions, with density distributions that have similar shapes within the data type. Tier 2 data

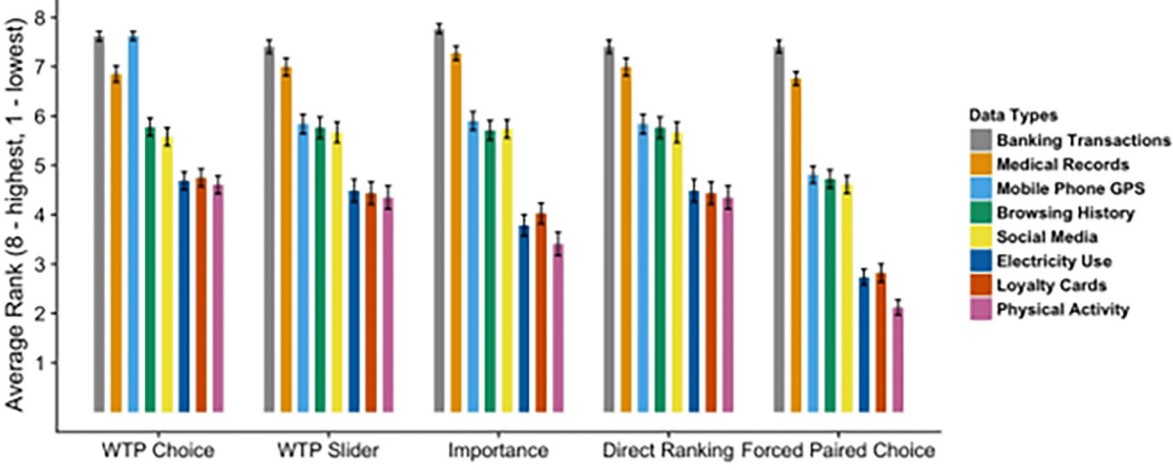

**Fig 3. Average ranking and corresponding 95% Confidence Intervals for all conditions.**

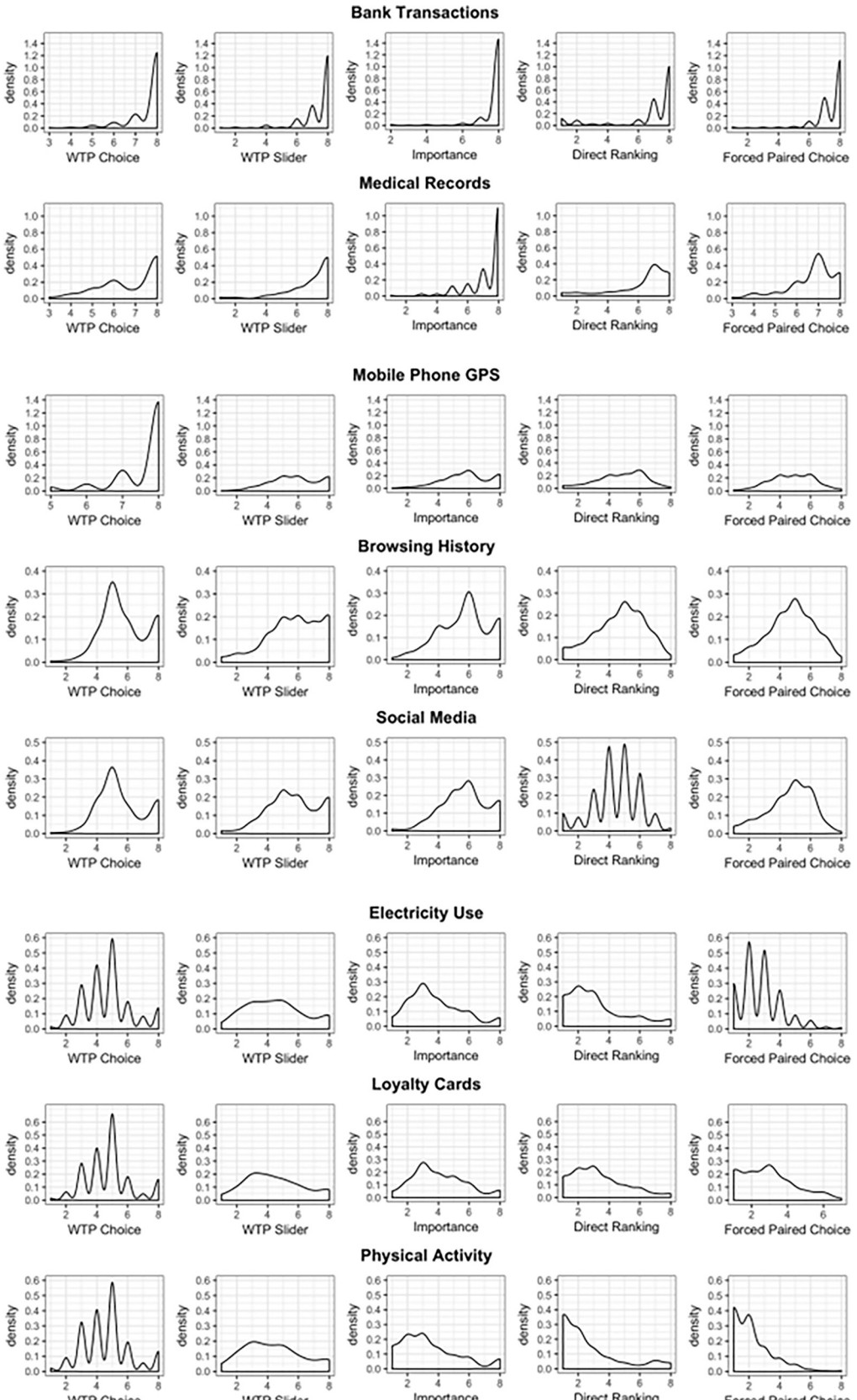

**Fig 4.** a. Density distribution of ranks for Tier 1 data: Banking and Medical Records Data. b. Density distribution of ranks for Tier 2 data: Browsing History, Mobile Phone GPS and Social Media Data. c. Density distribution of ranks for Tier 3 data: Electricity Use, Loyalty Cards and Physical Activity Data.

types have more variation and Tier 3 types have the least stability in ratings between conditions which is observable from different shapes of density distributions between conditions for the same data type. In other words, for Tier 1, and to a certain extent for Tier 2, it did not matter which elicitation procedure was used to elicit the ranks of the personal data types; the implied rankings were stable and consistent between conditions. However, for Tier 3, the shape of distribution seems to be mostly driven by the condition, not by data type: the distributions are more similar reading up the columns than along the rows for these data types. This can be interpreted to mean that the preferences with respect to Loyalty Cards, Electricity Use and Physical Activity data were less well defined, and hence the responses were driven more by the elicitation procedure than by true underlying privacy preferences.

**Individual level consistency.** For each participant we calculated Krippendorff's alpha coefficient to reflect how consistent they were in ranking different data types. The average Krippendorff's alpha coefficient was 0.51 (median 0.60, standard deviation of 0.29), ranging from -0.17 to 0.92. For the for the degree of consistency implied by these coefficients, see rankings provided by a randomly selected participant from the bottom 5th percentile of the consistency distribution, and by a randomly selected participant in the top 5th percentile in the S2 Appendix. To describe the whole sample in terms of consistency, we adopted the benchmark scales suggested by Landis and Koch [44, see 45] which allows participants to be classified in terms of their consistency scores. The share of participants who display poor, slight, fair, moderate, substantial and almost perfect consistency are shown in Table 1. The table demonstrates that only 20.6% of participants are below the threshold for "fair" consistency with the majority (70.61%) being moderately, substantially or almost perfectly consistent between conditions. To describe the whole sample in terms of consistency, we adopted the benchmark scales suggested by Landis and Koch [44, see 45] which allows participants to be classified in terms of their consistency scores. The share of participants who display poor, slight, fair, moderate, substantial and almost perfect consistency are shown in Table 1. The table demonstrates that only 20.6% of participants are below the threshold for "fair" consistency with the majority (70.61%) being moderately, substantially or almost perfectly consistent between conditions.

**To pay or not to pay?** 96% of participants stated for at least one data type that they would pay to protect it. There were differences in the likelihood with which individuals would be willing to pay for different data types. Fig 5 displays the results of a Probit regression predicting whether participant were willing to pay to protect their personal data, as a function of data types, consistency and demographic variables. We used a Mixed-model Probit (predicting willingness to pay) based on the Choice condition and the Slider condition, Full regression output is in S3 Appendix.

**Table 1. Break down of Krippendorff alpha consistency scores by poor, slight, fair, moderate, substantial and almost perfectly consistent.**

| Coefficient | Interpretation | Number of Participants | % of participants |
| --- | --- | --- | --- |
| <0 | Poor | 24 | 9.15% |
| [0.00–0.20] | Slight | 30 | 11.45% |
| [0.21–0.40] | Fair | 23 | 8.79% |
| [0.41–0.60] | Moderate | 52 | 19.85% |
| [0.61–0.80] | Substantial | 106 | 40.46% |
| [0.81–1.00] | Almost Perfect | 27 | 10.30% |

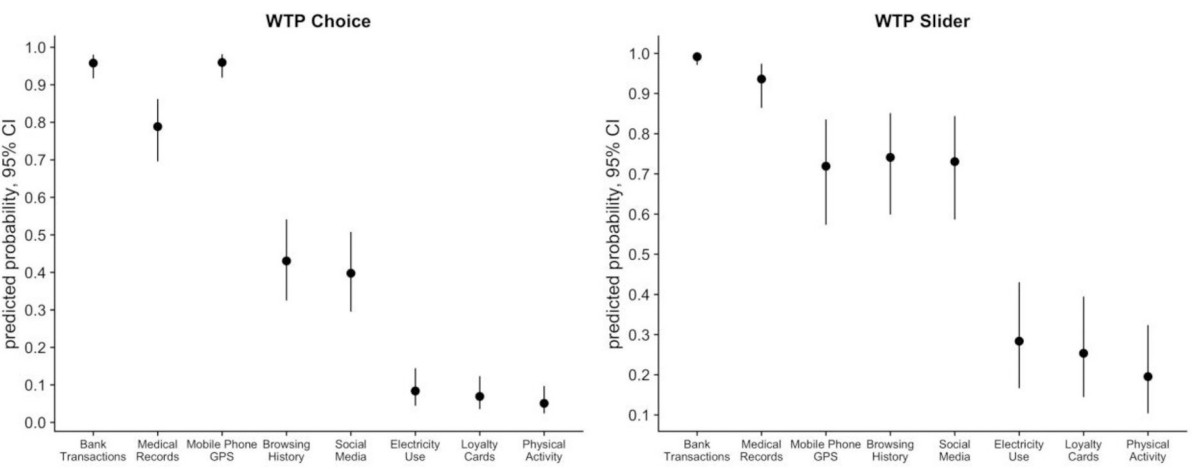

**Fig 5. Predicted probability to pay for different data types, 95% Confidence Intervals.**

The figure demonstrates predicted effects for each data type, conditioned on all other variables. The data type that was considered the most important to protect was Banking Transactions, with an estimated likelihood of being willing to pay to protect at 95.8% in WTP Choice and 99.2% in WTP Slider conditions, respectively. For Medical Records, the estimated likelihood of being willing to pay to protect was 78.9% in WTP Choice and 93.6% in the WTP Slider conditions, respectively. The second tier of data are Browsing History (43% in WTP Choice and 74.1% in WTP Slider) and Social Media (39.8% in WTP Choice and 73.1% in WTP Slider). Mobile Phone GPS was closer to the first tier in the WTP Choice condition with 96% predicted probability to pay, while in WTP Slider condition it was less protected: 71.9%. Finally, the third tier included Loyalty Cards (6.9%, WTP Choice; 25.3%, WTP Slider), Electricity Use (8.4%, WTP Choice; 28.4%, WTP Slider) and Physical Activity (5.1%, WTP Choice; 19.5% WTP Slider).

Individual consistency between ranking orders predicted higher likelihood of being willing to pay to protect personal data: those participants who demonstrated higher consistency in their responses were more likely to decide to pay something in both WTP conditions—that is, a one standard deviation increase in people's consistency measurement is estimated to cause a 9% and a 4% increase in the likelihood to pay to protect their personal data in the WTP Choice and WTP Slider conditions, respectively. This might reflect more thorough consideration of consequences of sharing personal data by those who showed higher consistency.

Out of the demographic variables, only age was negatively related to the likelihood to pay to keep the data private, and this was the case only in the WTP Slider condition: younger people were more willing to pay to keep their data private, with a one standard deviation (~11.42 years) decrease in age leading to a significant increase of 6% in the likelihood of being willing to pay to keep data private. Further, more educated participants, specifically those who stated that they held a postgraduate degree, were 10% more likely to choose to pay in WTP Choice condition. Income level had no significant effect on this decision.

See Table A2 in S3 Appendix for the regression output from the Mixed-model Probit (predicting whether participant were willing to pay for the relevant conditions.

**How much to pay?.** We have considered the results of the Probit analysis which investigated the factors influencing whether or not participants were willing to pay a non-zero amount to protect their personal data. Next, we turn to the OLS regression which investigates the level of WTP, conditional on being willing to pay an amount greater than zero. Fig 6

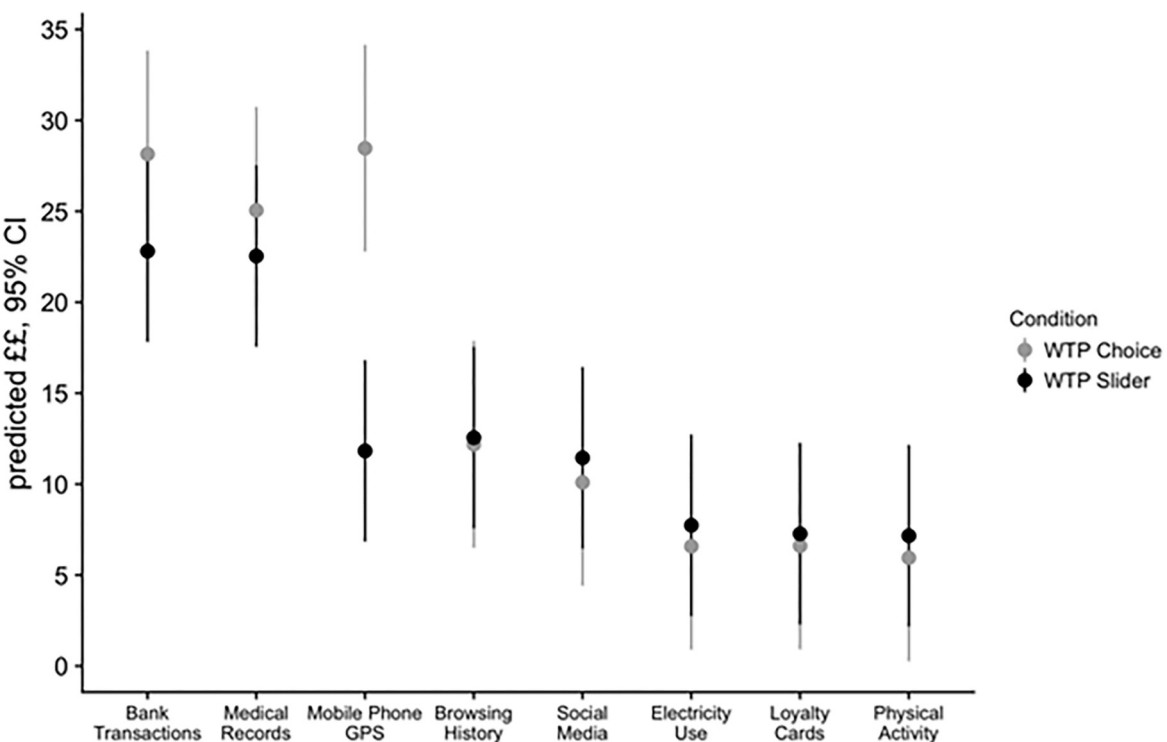

**Fig 6. Predicted amount individuals are willing to pay for different data types, out of those who were prepared to pay something, 95% Confidence Intervals.**

displays the predicted WTP amounts in pounds and the associated 95% Confidence Intervals for each data type based on the results of the OLS regression. These results differed from, but did not seriously contradict, the results of the Probit analysis. We discuss the OLS results and differences from the Probit below.

The results related to the differences between data types were consistent with those from the Probit analysis, suggesting—reassuringly- that the data types that people are more likely to pay to protect are also the data types that they are willing the pay largest amounts to protect. The estimated WTP to protect Bank Transaction data was £28.10 per month in the Choice condition and £22.80 in the Slider condition. The estimated WTP for protecting Medical Records was £25.10 for the Choice condition and £22.50 for the Slider condition. The estimated WTP for Slider and Choice conditions for the Tier 2 data types were as follows: Mobile Phone GPS £28.50 and £11.80; Browsing History £12.12 and £12.60; and Social Media £10.10 and £11.40, respectively. Finally, our participants were willing to pay £6.61 and £7.27 for Electricity Use, £6.58 and £7.74 for Loyalty Cards and £5.95 and £7.17 for Physical Activity data in the WTP Choice the WTP Slider condition, respectively.

Individual consistency in ranking order within condition was not associated with the amount of money participants were willing to pay in either condition. Similar to the decision about whether to pay anything at all, the amount that participants were willing to pay was associated negatively with age: one standard deviation decrease in years (~11.42 years) leads to an average payment to protect personal data that is £2.27 higher, but only in the WTP Slider condition. Female participants were willing to pay smaller amounts compared to men in both WTP conditions: £8.06 less in Choice and £5.13 less in Slider, on average. Finally, having a graduate degree was associated with being willing to pay more to protect data in both

conditions: £18.71 more in the Choice condition and £10.06 more in the Slider condition. As with the Probit, income did not have a significant effect on WTP. See Table A2 in S3 Appendix for the regression output OLS model predicting the amount people were WTP,

## Discussion

This paper makes three main contributions. First, we showed that individuals state they are willing to pay non-zero amounts to protect their personal data. Second, individuals demonstrate consistent rankings of the importance of protecting different types of their personal data, with values and ranks being remarkably similar across different elicitation methods. Third, different types of personal data are valued differently, suggesting there is no unified "price tag for privacy". Our findings reveal consistency in people's decisions about whether to share their personal data through a variety of elicitation procedures, which is evidence for consistent with the notion of stable preferences for privacy.

We now discuss how our findings relate to previous research exploring the value of personal data, explain how our methodology allowed us to unpack privacy and its value, and discuss how our results shed light on the possibilities for emerging business models and policy approaches in the domain of personal data protection. Different types of personal data might have different value for different stakeholders, and personal data may even have different value for the same consumer in various scenarios (e.g., a scenario in which a company provides services in exchange for consumer data vs a scenario in which consumers sell their data to company). Our study is the first to unpack the relative importance of protecting different types of personal data, ranging from very subjectively important, such as medical records and financial information, to—at the present time—subjectively unimportant, such as loyalty cards and electricity data. Our study deliberately did not control for specific risks and benefits of sharing as we aimed to mimic the real-world choices that people often face, where risks and benefits of agreeing to share one's data are typically unclear, although future research could usefully contribute to the literature by exploring these types of effect.

### Privacy preferences

Our work extends the growing literature on the monetary valuation of privacy [16, 46]. It also contributes to the body of knowledge concerning the factors that influence individuals' decisions about whether or not to share elements of their personal data online. To the best of our knowledge, this is the first report of a study that directly compares the value of protecting different types of personal data using five different elicitation techniques. We explored two main questions: whether individuals would state that they were willing to pay to protect different types of personal information, and whether their responses would imply rankings that were stable across different evaluation techniques, since economic theory suggests that if stable and well-defined privacy preferences exist, individuals' stated WTP would be consistent across different evaluation techniques.

In line with some previous research [18] and in contrast to other studies [46], our findings show that individuals state positive willingness to pay to ensure their personal data remains private, at least for some types of data. We observed a surprising degree of consistency in participants' rankings of different data types elicited through different techniques, which indicates that stated preference evaluation techniques may be appropriate tools to reveal individuals' underlying preferences for privacy. In our study, all factors that could affect people's decisions about what value they assign to different types of personal data remained the same, with only data types and evaluation techniques varying between the questions. We suggest that while the rankings implied by the responses in our study display remarkable consistency, there are still

features of the decision context that could influence the value of privacy online. These features may include the context of data sharing, the type of data that is shared, what the purpose of sharing the data is, and who the data is shared with. We suggest that these features will interact with underlying privacy preferences and contribute to producing the observed values of personal data. By not specifying or deliberately manipulating these features of the data sharing environment in our scenarios, we examined the effects of data type and elicitation procedure in isolation.

Our methodological approach and results demonstrate the benefits of clearly specifying privacy decision-making scenarios as choices about protecting specific types of data. The clear differences in value between different data types suggests that the value of privacy in the digital world cannot be defined as a unified concept: it is not possible to assign a single price tag to individual privacy. Instead, our findings suggest that in order to understand the value of privacy, it is necessary to study how individual privacy preferences play out in different data sharing environments. For example, in our study there was a clear contrast between very high rates of willingness to pay to protect banking transactions data and very low rates of willingness to pay to protect electricity use or physical activity data, demonstrating that public preferences for keeping the former type of data private are stronger than for the latter types. In addition, responses about low-valued data, referred to earlier in the paper as Tier 3, were noisier and appeared to be more strongly influenced by the elicitation techniques than for the high-valued Tier 1 and 2 data types. This result suggests that individuals have less well-defined preferences about protecting the Tier 3 data types than they do for the highly valued data types.

Understanding how much people value different types of personal data is an important first step in understanding individual privacy preferences and how they affect decisions in the real world. Everyday, people face scenarios where they are presented with requests to share different types of personal data, albeit the context might vary. Our results and the evaluation framework that we have introduced in this paper could be used to further investigate how those valuations change with different framing [e.g., similar to 47], or when the instructions manipulate the emphasis placed on the risks versus the benefits of sharing personal data online. However, we emphasise again that we are not advocating for the creation of markets for personal data in which individuals are required to pay to protect their privacy. Whilst our results suggest that at least some people would be willing to engage in such a market, the ethical considerations of this approach to managing online personal data are complex and the potential for exacerbation of inequality, and for the emergence of privacy as a luxury good, are troubling. Instead, by demonstrating that individuals' willingness to pay to protect their personal data is positive and relatively stable, we hope to have provided valuable information about the strength of public preferences over privacy, which can feed into the development of online protections, regulations, and business models that better reflect the preferences of the public. Essentially, by placing a monetary value on the protection of personal data, we provide an estimate for the benefit side of a benefit cost analysis of privacy protection.

## Limitations and future research

The choices in our experiments were hypothetical, following the tradition of the literature on the monetary valuation of non-market goods. We opted to take a hypothetical approach because it was not possible to identify real life choices (revealed preferences) in which different data types are traded in otherwise identical scenarios. Whilst incentivizing the sharing of personal data in experimental conditions is possible, and has been done before [e.g., 46], it was not appropriate for our research question because it is not possible to conduct an ethical experiment where individuals would be requested to pay money to avoid sharing their personal data

in a variety of highly sensitive data sharing environments. Using hypothetical stated preference tasks avoids this necessity by allowing us full control of the data sharing and payment scenarios and making it possible to ask about sensitive data types.

Furthermore, it is not currently feasible to construct an incentivised study where these different types of personal data are traded. Based on our findings, which showed remarkable consistency in implied rankings of and valuations for privacy, future research could be usefully conducted that focuses on a single type of data (e.g., retail loyalty cards) involving a field study where individuals could opt to pay for privacy or else opt to benefit from a free service. Building such a scenario would allow the researcher to manipulate different aspects of context of data sharing, to examine whether and how these aspects affect individual decisions. It would unlikely, however, to conduct such experiment with multiple different data types.

Decisions about sharing personal data are always contextual [3]. In our study we deliberately did not define context and reasons for sharing personal data. Future research could manipulate whether the data is shared for commercial purposes or for the public good to investigate whether this affects people's decisions about whether to share their personal data, and whether different reasons for sharing affect people in a different way.

We might also consider willingness to accept instead of willingness to pay: the literature on the economics of privacy and security implicitly assumes that the market behaviour of individuals for both actions is identical, but related literature in psychology suggests otherwise. Some previous research [e.g.,11] elicits individuals' willingness to accept money in return for sharing their personal data, instead of willingness to pay to protect it. This approach can underpin business models in which individuals have the right to privacy but can choose to sell their personal data, for example through companies like digi.me that give individuals the chance to directly choose how much data to share for financial reward. In contrast, we took a willingness to pay approach and found that individuals' willingness to pay to protect their data was positive. Indeed, since the introduction of the GDPR legislation in the EU, the status quo has shifted to one where the data subject has the right to control their data which might make a WTA framework more appropriate in future analyses. Nonetheless, a WTP approach is in line with most non-market valuation studies across contexts [48]. A final alternative, that might be explored in future research, is one in which consumers receive a specific, non-monetary benefit for sharing their data. Understanding the gap between willingness to pay and accept, as well as whether different scenarios affect values which are produced through different elicitation techniques, may have implications for future business models which evolve based on the use of personal data.

## Implications

Finally, our results have implications for policy-making. Currently, the digital economy does not typically provide a direct monetary benefit for consumers who create personal data. The benefits are instead provided through increased convenience of service delivery, or better opportunities for accessing certain services. Largely, this relies on the assumption that the actual value of one consumer's personal data is very hard to estimate and even if it is possible, the value is so small that consumers do not consider their personal data to be valuable enough to demand monetary compensation for sharing it. However, our findings, in line with previous research [see 3] suggest that there may exist an imbalance in power between the consumer and the data collector/holder in terms of the allocation of the welfare gain, whereby the consumer places a positive value on protecting their personal data and yet the welfare gain from the use of that personal data is allocated solely to the data handler.

Our results strongly suggest that privacy is not a unified concept and people make different decisions depending on what kind of data they are sharing. The types of detriment that are experienced when data are not protected can be diverse—including privacy loss *per se*, risks of adverse consequences for the individual, and a basic disapproval of situations where a private company makes money from one's own personal data. It is important to understand the preferences of members of the public towards all of these elements when making decisions about how to create and regulate a fair environment in which individuals are able to make informed choices about protecting their privacy. Similar to health economics where resources are distributed based on public evaluations of different health outcomes [2], arguably government regulations with respect to personal data and digital privacy should take account of the value of personal data privacy to society. Our study is the first to look at a breadth of different data types in terms of how they are evaluated by members of public, and we provide reassuring evidence in favour of the stability of preferences for online privacy. Our results therefore indicate how public preferences might be included in policy making in this important context.

## Supporting information

**S1 Appendix. Example of converting forced paired choices into a ranking.**
(DOCX)

**S2 Appendix. Two randomly selected participants from the top and bottom consistency scores distribution.**
(DOCX)

**S3 Appendix. Tabular format of regression output.**
(DOCX)

**S1 File.**
(DOCX)

## Author Contributions

**Conceptualization:** Anya Skatova, Rebecca McDonald, Carsten Maple.

**Data curation:** Anya Skatova, Rebecca McDonald.

**Formal analysis:** Anya Skatova, Rebecca McDonald, Sinong Ma.

**Funding acquisition:** Anya Skatova, Carsten Maple.

**Methodology:** Anya Skatova, Rebecca McDonald, Carsten Maple.

**Writing – original draft:** Anya Skatova, Rebecca McDonald, Sinong Ma, Carsten Maple.

**Writing – review & editing:** Anya Skatova, Rebecca McDonald, Sinong Ma, Carsten Maple.

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
