## [Decision Letter · Decision Letter 0]

2 Jan 2023

PONE-D-22-24992Unpacking Privacy: Willingness to Pay to Protect Personal DataPLOS ONE

Dear Dr. Skatova,

Thank you for submitting your manuscript to PLOS ONE. After careful consideration, we feel that it has merit but does not fully meet PLOS ONE’s publication criteria as it currently stands. Therefore, we invite you to submit a revised version of the manuscript that addresses the points raised during the review process.

I have received feedback on your work from two expert reviewers. We all see merit in your work, with potential to advance knowledge in a highly relevant field nowadays. However, referee 1 points to the ethical concern that a WTP to protect personal data approach raises, and referee 2 is not convinced about your stability of preferences argument. You must very carefully respond to these two issues if you decide to resubmit your work. Also a shortening of the paper is highly advisable. You must consider this eventual revision a high risk endeavour.

We look forward to receiving your revised manuscript.

Kind regards,

Iván Barreda-Tarrazona, PhD

Academic Editor

PLOS ONE

Journal Requirements:

Reviewers' comments:

Reviewer's Responses to Questions

**Comments to the Author**

1. Is the manuscript technically sound, and do the data support the conclusions?

Reviewer #1: Yes

Reviewer #2: Yes

2. Has the statistical analysis been performed appropriately and rigorously? 

Reviewer #1: Yes

Reviewer #2: Yes

3. Have the authors made all data underlying the findings in their manuscript fully available?

Reviewer #1: Yes

Reviewer #2: Yes

4. Is the manuscript presented in an intelligible fashion and written in standard English?

Reviewer #1: Yes

Reviewer #2: Yes

5. Review Comments to the Author

Reviewer #1: PONE-D-22-24992 “Unpacking Privacy: Willingness to Pay to Protect Personal Data”

This paper examines a pertinent socio-economic issue of data privacy in the digital world. The paper is well written. The empirical analysis is carefully done with the analysis of eight different types of data using five different methodologies. The results illustrate that individuals are willing to pay for data privacy and that people value data privacy differently for different types of data. The authors acknowledge the limitations of conducting hypothetical willingness-to-pay (WTP) or willingness-to-accept (WTA) surveys, a limitation that is well-recognized in the vast literature on non-market valuation.

I have a major concern about the approach of the paper and its potential implication. Why did the authors decide to analyze the WTP as opposed to WTA? The ethical implication of adopting WTP as opposed to WTA approach in the context of data privacy is serious. Is it ethical to propose a market where individuals have to pay for their data privacy? Prevalent economic inequality across the world would imply that if a market to pay for data privacy emerged, those in lower economic strata would be unable to protect their data due to their inability to pay, even if they highly value data privacy. Therefore, ethical considerations will suggest that since individuals ought to have complete ownership of their data, individuals need to be compensated if their data is used by someone else, especially if the data is used for commercial purposes. By adopting a WTP approach instead of the WTA approach, the property right over the data of the individuals is implicitly awarded to the private firms, and the individuals have to buy their right to protect the data. In an ethical society, it ought to be the opposite, wherein the default right to individual data lies with the individuals, and firms that would like to use the data need to pay the individuals for the data. To move towards a more ethical institutional setup, we need more conversation and policy interventions that push private firms toward ethical practices rather than promoting the idea that individuals have no default right to safeguard their data unless they pay for it. In sum, the potential policy implication of this paper that promotes the idea of making individuals pay for protecting their data is a serious concern from the perspective of socio-economic justice in human society. Unless the authors include an elaborate discussion of the ethical implications of their approach and highlight the potential pitfalls in the abstract, introduction and conclusion, I cannot recommend this paper for publication, especially in a high-impact journal like PLOS ONE whose core objective is to advance research for the benefit of society.

A minor correction: The round brackets are misplaced in the formula for α on page 17. The correct specification should be α = 1- (D0/De).

Reviewer #2: Conceptually, privacy is clearly a complex notion. Although in the introduction authors explain different properties and elements of privacy, a more structured conceptual framework of what affects/constitutes privacy would be useful. This would more clearly separate the preferences for privacy per se and confounders or consequences of privacy-related preferences, such as the willingness to share data. As it is, the paper offers a wide range of information

I am curious as to why the authors question whether individuals can consistently reveal their preference ordering with respect to the importance of protecting different types of their personal dana since they provide ample of evidence from other fields that this is not the case. What would make privacy – a complex and context-dependant notion – different, if anything? The authors conclude that stability in stated preferences for keeping personal data private across a variety of elicitation methods is evidence that privacy preferences are well defined. However, respondents provided valuations of keeping different types of data private which follow a particular order (highest value for keeping bank statement private, lowest for keeping physical activity private). Is that evidence of “stability” of preferences? We do not know which of these goods are more valuable to the market, there is no benchmark, so it is difficult to judge whether the ordering of preference is meaningful (a bigger orange relative to a smaller orange, to use the example the authors are using). The real test would be to measure the value of keeping private e.g., one bank statement vs 10 bank statements, or keeping data private for 1 year vs 10 years. What authors show is that people value the same good in a same order using different methods. There is no within-goods comparisons, only consistent ordering between very different types of goods. For instance, if respondents valued a house, a car and a cat in the same ordering using an open-ended WTP question and a slider, or a bidding game, and we found that they always value a house more than a cat, would that tell us that that the preferences are stable and therefore well-defined? What is meant by “well-defined”? I think a more thorough investigation and a different study design would be needed to make a claim about the “goodness” of preferences.

The paper is long, I would suggest shortening the text throughout, using more concise language. Also, information (such as the aim of the paper) should be presented at a single point in the paper to avoid repeating the text but also preventing confusion with respect to the aim of the paper. At times, it seems that this is a purely methodological exploration and at times it seems that the focus is on the valuing personal data. Hence, the paper would benefit from a more focused writing approach.

Can the authors present their experimental design in a table format? The text is long and it is not easily followed.

Can the authors place some of the tables/results in the appendix, to increase the readability of the paper? Again, this comes back to the point of the paper being too long.

6. PLOS authors have the option to publish the peer review history of their article (what does this mean?). If published, this will include your full peer review and any attached files.

Reviewer #1: No

Reviewer #2: No

---

## [Author Response · Author response to Decision Letter 0]

6 Feb 2023

Dear Professor Barreda-Tarrazona,

Thank you for the opportunity to revise our manuscript for resubmission to PloS One. We have taken on board the reviewers’ comments, made substantial changes to the presentation of the paper, shortened it, and clarified key concepts and terminology. We have also engaged with the fundamental concerns of Reviewer 1 both in this response and in the body of the paper itself. We are grateful to both reviewers for their positive comments and for their insightful and constructive criticism. We are confident that the paper has improved considerably as a result of the changes we have made, prompted by the reviewers’ insights. We include a point-by-point response to the reviewers in what follows.

We look forward to hearing from you in due course.

Yours sincerely, 

Dr Anya Skatova (on behalf of the authors)

Reviewer #1: 

"This paper examines a pertinent socio-economic issue of data privacy in the digital world. The paper is well written. The empirical analysis is carefully done with the analysis of eight different types of data using five different methodologies. The results illustrate that individuals are willing to pay for data privacy and that people value data privacy differently for different types of data. The authors acknowledge the limitations of conducting hypothetical willingness-to-pay (WTP) or willingness-to-accept (WTA) surveys, a limitation that is well-recognized in the vast literature on non-market valuation."

Thank you for your positive comments regarding the research question and the quality of our analysis and write up.

"I have a major concern about the approach of the paper and its potential implication. Why did the authors decide to analyze the WTP as opposed to WTA? The ethical implication of adopting WTP as opposed to WTA approach in the context of data privacy is serious. Is it ethical to propose a market where individuals have to pay for their data privacy? Prevalent economic inequality across the world would imply that if a market to pay for data privacy emerged, those in lower economic strata would be unable to protect their data due to their inability to pay, even if they highly value data privacy. Therefore, ethical considerations will suggest that since individuals ought to have complete ownership of their data, individuals need to be compensated if their data is used by someone else, especially if the data is used for commercial purposes. By adopting a WTP approach instead of the WTA approach, the property right over the data of the individuals is implicitly awarded to the private firms, and the individuals have to buy their right to protect the data. In an ethical society, it ought to be the opposite, wherein the default right to individual data lies with the individuals, and firms that would like to use the data need to pay the individuals for the data. To move towards a more ethical institutional setup, we need more conversation and policy interventions that push private firms toward ethical practices rather than promoting the idea that individuals have no default right to safeguard their data unless they pay for it. In sum, the potential policy implication of this paper that promotes the idea of making individuals pay for protecting their data is a serious concern from the perspective of socio-economic justice in human society. Unless the authors include an elaborate discussion of the ethical implications of their approach and highlight the potential pitfalls in the abstract, introduction and conclusion, I cannot recommend this paper for publication, especially in a high-impact journal like PLOS ONE whose core objective is to advance research for the benefit of society."

Thank you for this interesting and important point. We agree entirely with your assessment of the dangerous ethical implications of advocating a world in which online privacy must be paid for. However, we do not agree that using WTP is the same as advocating for such a market to exist in reality. Instead, we use WTP (and could equally have used WTA) as a tool to elicit the value that individuals place on protecting their personal data. This value is then able to be fed into regulatory efforts, particularly via cost benefit analysis, that aim to protect online privacy. Indeed, if we draw a parallel with well-established health economics applications of the WTP methodology - in the UK WTP-based values of health and safety are used to feed in to policy evaluations of the benefit to society of improving health and safety, and this is not taken to imply that health and/or safety ought to be paid for in an open market. 

We have conducted a thorough review of our manuscript and removed any passages that may be inadvertently interpreted as advocating for such a market for personal data and we now include an explicit discussion of these issues both in the introduction (Page 8-9, Lines 199-207) and again in the conclusion (Page 29, Lines 651-661). We also expand our discussion of the decision to use WTP instead of WTA (Page 30-31, Lines 695-698). We remove references to WTP in the abstract, talking instead in broader terms about valuation, in order to avoid inadvertently creating the wrong impression up front.

"A minor correction: The round brackets are misplaced in the formula for α on page 17. The correct specification should be α = 1- (D0/De)."

We have corrected this typo in the manuscript.

Reviewer #2: 

"Conceptually, privacy is clearly a complex notion. Although in the introduction authors explain different properties and elements of privacy, a more structured conceptual framework of what affects/constitutes privacy would be useful. This would more clearly separate the preferences for privacy per se and confounders or consequences of privacy-related preferences, such as the willingness to share data. As it is, the paper offers a wide range of information"

This is a good point. We have included a paragraph in the introduction (Page 9, Lines 214-225) to clarify what we intend by the different concepts and on how we believe they relate to one another. This is supported by a new figure (Figure 1, page 10) which clarifies these links visually. We have also tightened up the wording throughout the paper, especially in the empirical part and the discussion, to ensure clarity on this point.

"I am curious as to why the authors question whether individuals can consistently reveal their preference ordering with respect to the importance of protecting different types of their personal dana since they provide ample of evidence from other fields that this is not the case. What would make privacy – a complex and context-dependant notion – different, if anything? "

The reviewer’s point is interesting, but we do not feel that the evidence from other fields about consistent preference revelation is entirely sewn up. Indeed, the non-market valuation literature in environmental economics, health economics and related fields is very well established, despite growing evidence of preference instability in these fields. Our paper provides an empirical test of whether the revealed preference ordering of data types is sufficiently stable to allow us to meaningfully infer the relative importance of different types of personal data - and we argue that our results suggest it is the case. Moreover, whilst we agree privacy is a complex and context-dependent notion, we would argue that online privacy decisions are encountered more regularly than decisions about health or the environment, and hence this application may be considered a priori to be a good candidate for this type of contingent valuation study, as indeed we find.

"The authors conclude that stability in stated preferences for keeping personal data private across a variety of elicitation methods is evidence that privacy preferences are well defined. However, respondents provided valuations of keeping different types of data private which follow a particular order (highest value for keeping bank statement private, lowest for keeping physical activity private). Is that evidence of “stability” of preferences? We do not know which of these goods are more valuable to the market, there is no benchmark, so it is difficult to judge whether the ordering of preference is meaningful (a bigger orange relative to a smaller orange, to use the example the authors are using). "

Thank you for opening this discussion. We acknowledge that revealing a stable ordering of options is not the same as revealing stable underlying preferences (which are unobservable), although of course it is evidence compatible with the existence of stable preferences as we claim in the paper. Essentially, our ranking stability is a necessary but not sufficient condition for claiming stable underlying preferences. We entirely agree that where the value of two items (in our case datatype privacies, corresponding to your cat vs house example below) are very different, then even if these values are very imprecisely defined, the ranking between them will remain stable. It is plausible that the difference in value of protecting data types across our three tiers conforms with this interpretation. However, considering the WTP conditions in particular to help us address this, we note that the 95% confidence intervals around these values are relatively tight and that the monetary values are not extremely far apart. This suggests that we do observe relatively stable rank orderings, at least between the tiers. Moreover, we believe that our use of Krippendorff’s alpha helps to bolster our claims of stability. That measure is a well-established means to test the stability of rankings between different raters (in our case, different elicitation methods) and we do not believe there is a clearly superior approach that we could have used. 

We have amended the draft to reflect a more cautious interpretation, which is now made in multiple places in the draft including the abstract. We now refer to our evidence as being “consistent with” the idea of stable underlying preferences (e.g, Abstract, Line 34).

"The real test would be to measure the value of keeping private e.g., one bank statement vs 10 bank statements, or keeping data private for 1 year vs 10 years. What authors show is that people value the same good in a same order using different methods. There is no within-goods comparisons, only consistent ordering between very different types of goods. For instance, if respondents valued a house, a car and a cat in the same ordering using an open-ended WTP question and a slider, or a bidding game, and we found that they always value a house more than a cat, would that tell us that that the preferences are stable and therefore well-defined? What is meant by “well-defined”? I think a more thorough investigation and a different study design would be needed to make a claim about the “goodness” of preferences."

We like this proposed design and would consider it a valuable approach to try in future studies. However, we do not think it is as simple as assuming 1 bank statement vs 10 bank statements would be valued in a ratio of 1:10 (perhaps much of the privacy loss from sharing 10 is already apparent when sharing a single one). In fact, this approach may also be criticised for being too open to experimenter demand (“I think I’m expected to rank 10 as worse than 1, so that’s what I’ll do”) whereas our design allows participants to reveal their values and rankings of options without a “right answer” - which we consider a strength rather than a drawback of our approach. Nonetheless, we are open to your point that there are potentially many different approaches for tackling our research questions and that some will have advantages. We consider this fruitful ground for progressing the research agenda.

"The paper is long, I would suggest shortening the text throughout, using more concise language. "

We agree and have made efforts to shorten the text throughout. 

"Also, information (such as the aim of the paper) should be presented at a single point in the paper to avoid repeating the text but also preventing confusion with respect to the aim of the paper. At times, it seems that this is a purely methodological exploration and at times it seems that the focus is on the valuing personal data. Hence, the paper would benefit from a more focused writing approach."

We have checked for consistency of the write up paying particular attention to the stated aims of the paper.

"Can the authors present their experimental design in a table format? The text is long and it is not easily followed."

We did not manage to come up with a table format that could clearly explain our design, but we have shortened the text, included headers and included an additional figure aimed at clarifying the experimental design. We also make reference to the full experimental design details provided in Supplementary Materials.

"Can the authors place some of the tables/results in the appendix, to increase the readability of the paper? Again, this comes back to the point of the paper being too long."

We have taken this suggestion and moved some of our lengthier results to an appendix, as well as one of the supporting tables from the explanation of the analytic approach (see Appendices).

---

## [Decision Letter · Decision Letter 1]

29 Mar 2023

PONE-D-22-24992R1Unpacking Privacy: Willingness to Pay to Protect Personal DataPLOS ONE

Dear Dr. Skatova,

Thank you for submitting your manuscript to PLOS ONE. After careful consideration, we feel that it has merit but does not fully meet PLOS ONE’s publication criteria as it currently stands. Therefore, we invite you to submit a revised version of the manuscript that addresses the points raised during the review process.

Your manuscript has successfully incorporated most of the reviewers suggestions. Just a few points remain that you have to correct so as the paper to be publishable. Please respond to the points raised by the reviewer and make sure to incorporate the required modifications. Please submit your revised manuscript by May 13 2023 11:59PM. If you will need more time than this to complete your revisions, please reply to this message or contact the journal office at plosone@plos.org. Please include the following items when submitting your revised manuscript:A rebuttal letter that responds to each point raised by the academic editor and reviewer(s). You should upload this letter as a separate file labeled 'Response to Reviewers'.A marked-up copy of your manuscript that highlights changes made to the original version. You should upload this as a separate file labeled 'Revised Manuscript with Track Changes'.An unmarked version of your revised paper without tracked changes. You should upload this as a separate file labeled 'Manuscript'.If applicable, we recommend that you deposit your laboratory protocols in protocols.io to enhance the reproducibility of your results. Protocols.io assigns your protocol its own identifier (DOI) so that it can be cited independently in the future. For instructions see: https://journals.plos.org/plosone/s/submission-guidelines#loc-laboratory-protocols. Additionally, PLOS ONE offers an option for publishing peer-reviewed Lab Protocol articles, which describe protocols hosted on protocols.io. Read more information on sharing protocols at https://plos.org/protocols?utm_medium=editorial-email&utm_source=authorletters&utm_campaign=protocols.

We look forward to receiving your revised manuscript.

Kind regards,

Iván Barreda-Tarrazona, PhD

Academic Editor

PLOS ONE

Journal Requirements:

Reviewers' comments:

Reviewer's Responses to Questions

**Comments to the Author**

1. If the authors have adequately addressed your comments raised in a previous round of review and you feel that this manuscript is now acceptable for publication, you may indicate that here to bypass the “Comments to the Author” section, enter your conflict of interest statement in the “Confidential to Editor” section, and submit your "Accept" recommendation.

Reviewer #1: (No Response)

2. Is the manuscript technically sound, and do the data support the conclusions?

Reviewer #1: (No Response)

3. Has the statistical analysis been performed appropriately and rigorously? 

Reviewer #1: (No Response)

4. Have the authors made all data underlying the findings in their manuscript fully available?

Reviewer #1: (No Response)

5. Is the manuscript presented in an intelligible fashion and written in standard English?

Reviewer #1: (No Response)

6. Review Comments to the Author

Reviewer #1: (No Response)

7. PLOS authors have the option to publish the peer review history of their article (what does this mean?). If published, this will include your full peer review and any attached files.

Reviewer #1: No

---

## [Author Response · Author response to Decision Letter 1]

31 Mar 2023

Dear Professor Barreda-Tarrazona,

Thank you for the opportunity to revise our manuscript for resubmission to PloS One. We have taken on board the reviewer comments and made changes to the title, which now reads: “Unpacking Privacy: Valuation of Personal Data Protection” and one sentence in the abstract that they pointed out, which now reads: 

“One technique to assess how much individuals value their privacy is to ask them whether they might be willing to pay for an otherwise free service if paying allowed them to avoid sharing personal data.”

We hope this addresses their comments and look forward to hearing from you in due course.

Yours sincerely, 

Dr Anya Skatova (on behalf of the authors)

---

## [Editor Report · Decision Letter 2]

4 Apr 2023

Unpacking Privacy: Valuation of Personal Data Protection

PONE-D-22-24992R2

Dear Dr. Skatova,

We’re pleased to inform you that your manuscript has been judged scientifically suitable for publication and will be formally accepted for publication once it meets all outstanding technical requirements.

Kind regards,

Iván Barreda-Tarrazona, PhD

Academic Editor

PLOS ONE
---

## [Editor Report · Acceptance letter]

12 Apr 2023

PONE-D-22-24992R2 

Unpacking Privacy: Valuation of Personal Data Protection 

Dear Dr. Skatova:

I'm pleased to inform you that your manuscript has been deemed suitable for publication in PLOS ONE. Congratulations! Your manuscript is now with our production department. 

Kind regards, 

on behalf of

Dr. Iván Barreda-Tarrazona 

Academic Editor

PLOS ONE